# A SPACE-CONTINUOUS IMPLEMENTATION OF PROPER ORTHOGONAL DECOMPOSITION BY MEANS OF NEURAL NETWORKS

## ABSTRACT

In the realm of reduced order modeling, the Proper Orthogonal Decomposition (POD) has established itself as a widely adopted technique for efficiently handling parametric partial differential equations. This approach exploits principles of linear algebra to extract, from a collection of high-fidelity numerical solutions, an optimized reduced space capable of linearly representing the input data. This paper aims to introduce an innovative alternative to replicate the capabilities of POD by harnessing the power of neural networks, thereby overcoming the constraint of exclusively working with solutions confined to the same topological space. Our method centers around the utilization of the DeepONet architecture, which is applied and minimally modified to emulate the POD spatial-temporal (or parametric) decomposition. This novel adaptation enables the creation of a continuous representation of spatial modes. Although the accuracy gap between neural networks and linear algebraic tools is still evident, this architecture exhibits a distinct advantage: it can accept solutions generated through different discretization schemes, contrary to the conventional POD approach. Furthermore, our approach allows various enhancements and variants developed to augment the capabilities of POD. These can be seamlessly integrated into the architecture, offering a versatile and adaptable framework known as PODNet.

To validate its effectiveness, we apply it to two distinct test cases: a simple 1D trigonometric problem and a more complex 2-dimensional Graetz problem. In doing so, we conduct a comprehensive comparison between our proposed methodology and established approaches, shedding light on the potential advantages and trade-offs inherent to this innovative fusion of neural networks and traditional reduced order modeling techniques.

## 1 INTRODUCTION AND MOTIVATIONS

Proper orthogonal decomposition (POD) has become one of the most widespread techniques in the reduced order modeling (ROM) community Benner et al. (2021a;b;c); Rozza et al. (2022); Volkwein (2013). The method consists of the identification of recurrent patterns among the solutions of parametric partial differential equations (PDEs), in order to individuate the optimal low-dimensional space to represent the problem at hand linearly. Indeed, these kinds of equations require in most cases a massive computational power in order to be solved with the consolidated discrete techniques — finite element and finite volume — especially if the solution needs to be computed several times, for several parameters, with high accuracy.

POD alleviates the computational burden by reducing the dimensions of the original problem, providing an efficient model that mimics the behavior of the original (numerical) model in an almost real-time fashion. In order to reach this result, POD needs a limited set of high-fidelity numerical solutions[1], the so-called snapshots, precomputed and arranged as columns of a matrix. Singular value decomposition is then applied to the latter to compute the reduced space's basis. Finally, this space is exploited in a Galerkin framework Morelli et al. (2023); Buoso et al. (2019) or to interpolate a low dimensionality representation of the original snapshots Tezzele et al. (2022); Guo & Hesthaven

---

[1]corresponding to a fixed set of parameters

(2019); Hesthaven & Ubbiali (2018), in order to compute new solutions for new parameters. During the last years, several extensions of the POD have been presented, tackling the well-known limitations of the technique Fresca & Manzoni (2022); Lee & Carlberg (2020); Coscia et al. (2023). These approaches involve novel, sophisticated techniques inherited by the machine learning field in order to fight the disadvantages of the standard one. Most of the proposed solutions show promising results, even if replacing (or appending to) the POD with neural networks a loosing consequently the nice features of the original method.

However, many of these proposed solutions, while promising, often involve replacing or augmenting POD with neural networks, resulting in the loss of some of the favorable attributes of the original method. This contribution deviates from proposing a direct extension of the existing framework and, instead, focuses on casting the POD formulation to the context of neural networks. We name the presented methodology PODNet. The goal is to explore the use of neural networks for learning the distribution of POD modes and the reduced solution manifold. In this context, the POD modes are no longer confined to a specific discrete space, rendering them more versatile and suitable for use with heterogeneous data sources, including numerical solutions defined on different spaces, experimental data, and sensor data.

While it is challenging to surpass the accuracy of traditional POD through neural network training, this endeavor opens up several promising avenues for future research:

- With the continuous representation of spatial modes, the PODNet can provide predictions for any unseen parameters and coordinates within the parameter and spatial domains without the need for additional interpolation.
- Numerical solutions from diverse spatial domains, all representing the same problem, can be effortlessly integrated during training, along with other numerical measurements, including experimental data.
- Unlike other non-intrusive POD-based approaches, PODNet simultaneously optimizes the mapping to the reduced space and the approximation of the solution manifold. This cohesive approach enhances generalization and mitigates potential stability issues.
- Leveraging the continuous modes, numerous extensions originally developed for standard POD can be seamlessly transferred to this proposed approach, including the application of physics-informed paradigms Raissi et al. (2019) and mode transformations.

The contribution introduces the details of the PODNet architecture in Section 2, while in Section 3 we present the numerical results of the PODNet investigation on two different problems with incremental complexity. Finally, Section 4 summarizes the authors' claim and provides new research perspectives, hopefully enabled by the present contribution.

## 2 THE PODNET

POD is based on the spatial-parametric[2] division. Formally, a generic scalar parametric field $u(x; \mu)$ for $x \in \Omega \subset \mathbb{R}^d$ and $\mu \in P \subset \mathbb{R}^p$ can be represented by infinity sum:

$$u(x; \mu) = \sum_{i=1}^{\infty} \phi_i(x) \alpha_i(\mu), \tag{1}$$

which can be truncated to the first $k$ terms to obtain a good enough approximation Volkwein (2013).

The spatial structures $\{\phi_i(x)\}_{i=1}^k$ are the POD modes, aka the basis functions of the reduced space. The latter is computed by processing a limited set of high-fidelity solutions, typically obtained by employing consolidated approaches (finite element, finite volume to cite two examples). The database of solutions, arranged by matrix, is processed by linear algebra technique, like singular value decomposition, to find the modes. In the standard approach, the latter belongs to the same space where the high-fidelity solutions are computed (which needs to be unique), making the entire reduced-order modeling somehow anchored to a specific space.

PODNet implements instead the Eq 1 using neural networks to represent the $\phi_i : \mathbb{R}^d \to \mathbb{R}$ and $\alpha_j : \mathbb{R}^p \to \mathbb{R}$, where $d$ and $p$ are the spatial and parameter dimensions, respectively. Such networks, formally ModeNet and CoefficientNet, are trained all at once, using the input snapshots for

---

[2]The time can be here considered as a parameter

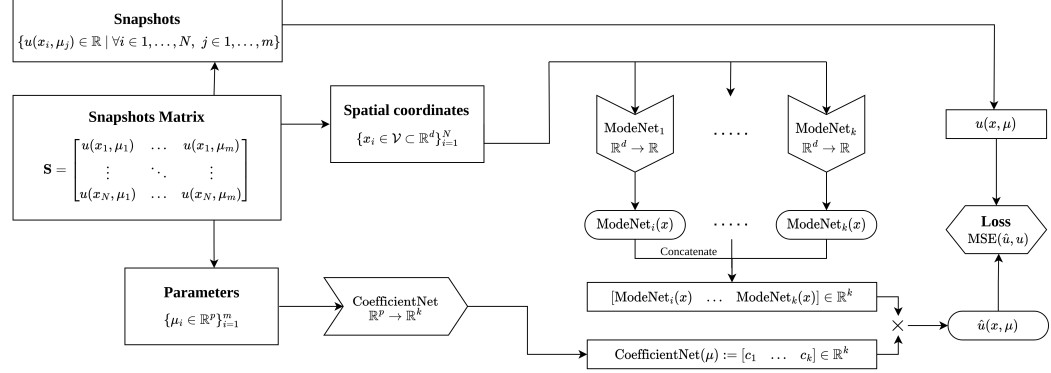

Figure 1: The PODNet structure.

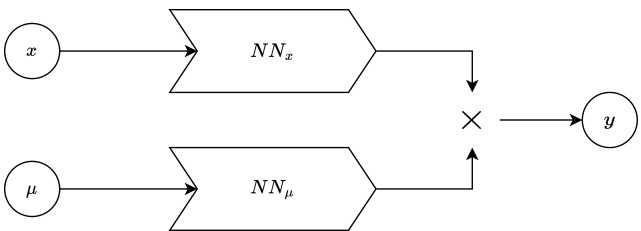

Figure 2: A simplified view of the DeepONet architecture.

the loss computation, as sketched in Figure 1. The final prediction is so obtained by computing $\hat{u}(x, \mu) = \sum_{i=1}^{k} \text{ModeNet}_i(x) \, \text{CoefficientNet}(\mu)_i$, whereas its distance with respect to the high-fidelity data is used for the loss computation. In the current contribution, standard mean squared error is employed, but the model can be surely trained also using different metrics.

## 2.1 ARCHITECTURE

The formal architecture is a (stacked) DeepONet Lu et al. (2021). While this architecture was initially designed to learn nonlinear operators, in our current work, its primary objective is to emulate the behavior of POD. Essentially, we are using it to model a parametric operator that operates on the output field. From a practical perspective, DeepONet is composed of two networks taking different components — e.g. the spatial coordinates and the parameters — of the input, as shown in the summary scheme at Figure 2. The outputs of the networks are subsequently multiplied element-wise, and the sum of these multiplications yields the final output. In this context, a stacked architecture is involved, preferring to have $k$ independent networks instead of a unique larger one which needs to learn all the modes of distribution.

## 2.2 ORTHOGONALITY CONSTRAINTS

One crucial feature of the POD technique is the capability to extract orthogonal basis, enabling the possibility to use them in a projection framework. In order to obtain a similar behavior using the PODNet, we added a new term in the loss function, following the idea in Kim & Yun (2022).

$$\mathcal{L}_{\text{ortho}} = \lambda \|\mathbf{W}\mathbf{W}^T - I\|_2^2, \quad \mathbf{W} = \begin{bmatrix} \text{ModeNet}_1(x_1) & \dots & \text{ModeNet}_1(x_n) \\ \vdots & \ddots & \vdots \\ \text{ModeNet}_k(x_1) & \dots & \text{ModeNet}_k(x_n) \end{bmatrix}, \quad (2)$$

where $\mathbf{W} \in \mathbb{R}^{n \times k}$ contains the evaluation ad the $k$ ModeNets in $n$ sample points (arranged by row), and $I$ refers to identity matrix. The idea to efficiently compute the orthogonality coefficient between the modes is to evaluate such networks on the $\{x_i\}_{i=1}^{n}$ testing points, obtained by sampling the spatial domain. It is not mandatory to use the same spatial coordinates of the input snapshots,

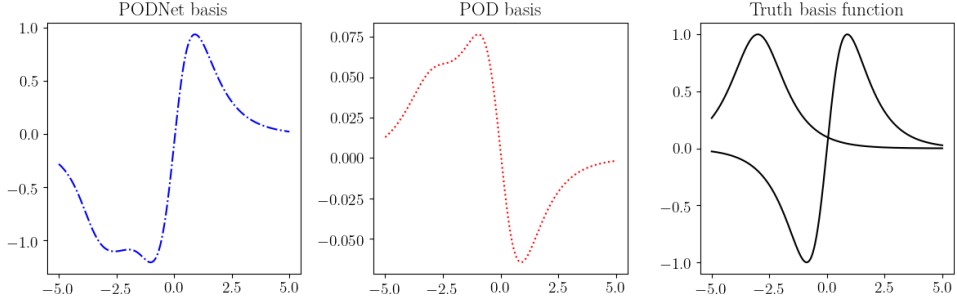

Figure 3: *Trigonometric Problem*: the spatial mode computed by using standard POD approach and using PODNet ($k = 1$).

but they can be arbitrarily chosen. The parameter $\lambda \in \mathbb{R}^+$ acts as a sort of Lagrangian multiplier, helping the training convergence since the weak imposition of the orthogonality.

## 3  NUMERICAL RESULTS

This section is dedicated to showing the results obtained by applying the PODNet on two different test cases, comparing them to POD-based methodology: POD with radial basis function (POD-RBF), and POD with artificial neural network (POD-ANN).

### 3.1  TRIGONOMETRIC PROBLEM

In the first experiment, the high-fidelity solutions are generated by the sum of two trigonometric functions evolving in time. Formally:

$$u(x,t) = f_1(x,t) + f_2(x,t), \quad \text{where} \begin{cases} f_1(x,t) = \text{sech}(x+3)\exp(i2.3t) \\ f_2(x,t) = 2\text{sech}(x)\text{tanh}(x)\exp(i2.8t) \end{cases} \tag{3}$$

The solution database is constituted by the sum of the two functions evaluated in $512$ spatial points, equispaced in the interval $[-5, 5]$, in $128$ time instants ($t \in [0, 4\pi]$). It must be said that the solutions we collect are the real part of the field of interest. We apply to this test case the standard POD, by arranging them as columns of a matrix, and the PODNet. For this experiment, the ModeNets are multi-layer perceptron (MLP) composed of 3 hidden layers of 25, 15, and 10 neurons. The activation function is the hyperbolic tangent. The CoefficientNet is still an MLP, but with 18, 18, and 8 neurons in the three hidden layers, and applying Softplus as the activation function. The training is performed using the Adam optimizer Kingma & Ba (2014) for 10000 epochs and the LBGFS optimizer for other 1000 epochs, in a cascade fashion.

**Reduced dimension $k$ = 1.**  We begin with a reduced dimension equal to 1. While this dimension is insufficient to fully represent the original system, our aim is to compare the performance when the reduced space lacks an adequate number of dimensions. Interestingly, both approaches yield highly similar single modes, with PODNet effectively capturing the inverse of the POD mode (see Figure 3). This symmetrical behavior and the differing scales are not problematic, as the multiplicative coefficient accounts for these distinctions.

Looking indeed to the test error in Figure 4, computed on new time instants, we can note that PODNet shows similar accuracy of POD-RBF, demonstrating that the approach can actually mimic the standard behavior.

**Reduced dimension $k$ = 2.**  We raise then the reduced dimension to 2. We repeat the same experiment and identify the modes using the approaches. It is important to note that here we start to use the orthogonality weight $\lambda = 1.0$ in order to enforce the mode to be orthogonal. Figure **??** shows the modes, again showing that the proposed network is able to replicate the expected trend.

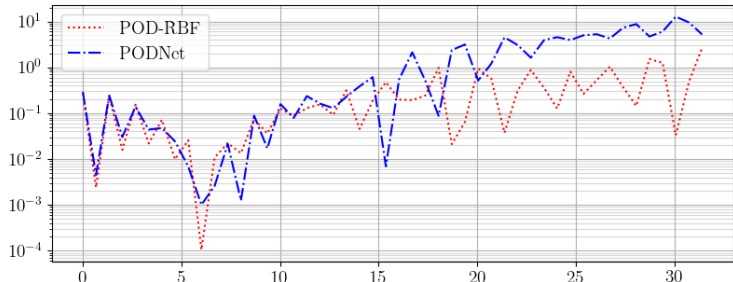

Figure 4: *Trigonometric Problem*: the mean error at different test time instant by using standard POD approach and using PODNet ($k = 2$), using an orthogonality weight $\lambda = 1$.

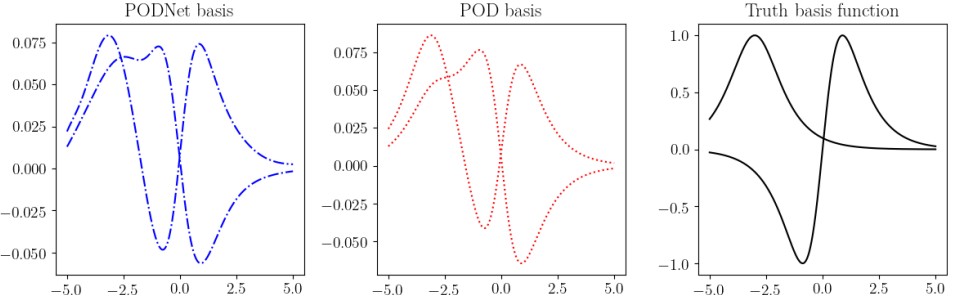

Figure 5: *Trigonometric Problem*: the spatial mode computed by using standard POD approach and using PODNet ($k = 2$), using an orthogonality weight $\lambda = 1$.

Trying instead to reproduce the same test, but without imposing the modes orthogonality, we noted the capability of PODNet to identify the basis function that originated the solutions. Figure 6 demonstrates the similarity between the original functions ($f_1$ and $f_2$) evaluated at time $t = 0$ and the computed modes, contrary to standard POD. Such difference does not alter the final accuracy, which is greater for the standard method, but demonstrates the importance of the orthogonality constraints.

## 3.2 2-DIMENSIONAL GRAETZ PROBLEM

The second test case deals with the Graetz-Poiseuille problem, which models forced heat convection in a channel. The problem contains 2 parameters: $c_1$ controls the length of the domain, while $c_2$ is the Péclet number, that takes into account the heat transfer in the domain. The full domain is $\Omega(c_1) = [0, 1 + c_1] \times [0, 1]$, with $\mathbf{c} = [c_1, c_2] \in [0.1, 10] \times [0.01, 10]$. The high fidelity solution

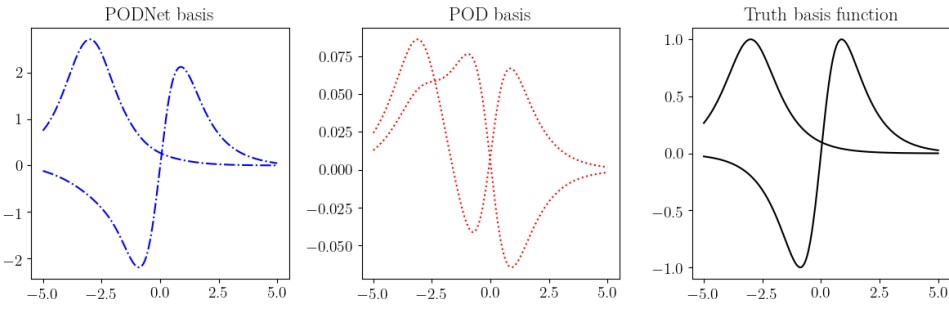

Figure 6: *Trigonometric Problem*: the spatial mode computed by using standard POD approach and using PODNet ($k = 2$), using an orthogonality weight $\lambda = 0$.

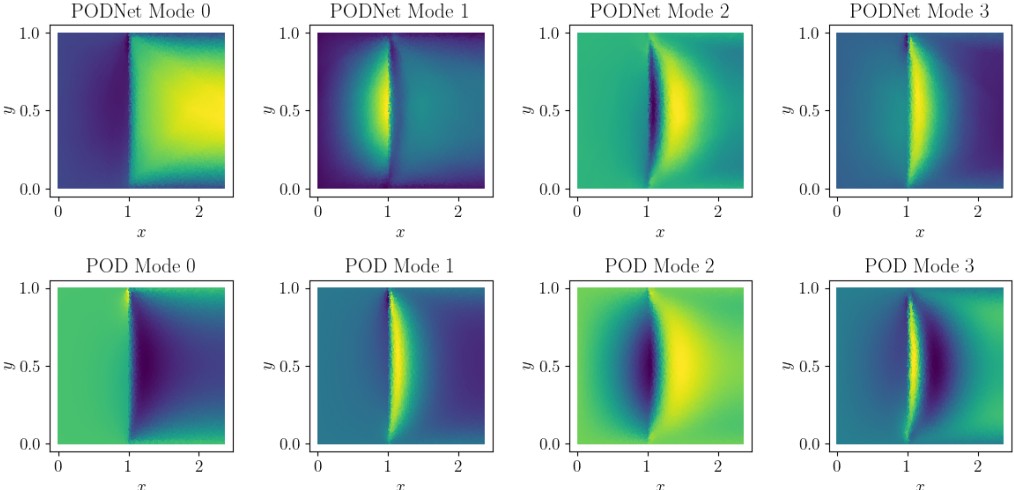

Figure 7: *Graetz Problem*: the 4 modes computed using standard POD approach and using PODNet.

$u : \Omega(c_1) \to \mathbb{R}$ is obtained by solving:

$$\begin{cases} -\Delta u(x, y; \mathbf{c}) + c_2 y(1 - y)\frac{\partial}{\partial x}u(x, y; \mathbf{c}) = 0 & (x, y) \in \mathring{\Omega}(c_1) \\ u(x = 0, y; \mathbf{c}) = 0 & y \in [0, 1] \\ u(x, y = 0; \mathbf{c}) = 0 & x \in [0, 1] \\ u(x, y = 1; \mathbf{c}) = 0 & x \in [0, 1] \\ u(x, y = 0; \mathbf{c}) = 1 & x \in [1, 1 + c_1] \\ u(x, y = 1; \mathbf{c}) = 1 & x \in [1, 1 + c_1] \\ \partial_\mathbf{n} u(x = 1 + c_1, y) = 0 & y \in [0, 1] \end{cases} \tag{4}$$

The high-fidelity solutions are numerically computed by finite elements following the work Hesthaven et al. (2016), using a mesh of 5160 points. The problem is solved for 200 parametric instances, uniformly sampled within the parametric domain, using 150 of them to train the models and the rest for the tests. We highlight that the final solution database contains all the numerical solutions projected onto a reference domain, to ensure standard POD applicability. For this experiment, the ModeNets are multi-layer perceptron (MLP) composed by 3 hidden layers of 40, 20, and 20 neurons. The CoefficientNet contains instead 20 and 10 neurons, arranged in two hidden layers. The training is performed using the Adam optimizer for 40000 epochs and the LBGFS optimizer for other 1000 epochs, in a cascade fashion. The reduced dimension is $k = 4$, whereas the orthogonality coefficient is here imposed to $\lambda = 0.0001$. Such value is manually tuned in a trial and error procedure since it affects the final loss of the model.

The first investigation, also in this case, wants to compare the spatial distribution obtained by using the two approaches. Even if the order is different — we remember that PODNet is not able to provide the energetic contributions of each mode and so the hierarchical order —, the distributions are quite similar (see Figure 7).

Regarding the precision, we compute the relative error using 50 test solutions. We compare three models: POD-ANN, POD-GPR, and PODNet. The network mapping the parameters on the modal coefficient in the POD-ANN is set equal to the CoefficientNet (in the PODNet) for a fair comparison. On average, PODNet is able to reach the same precision as POD-ANN, showing a more compact error distribution. Table 1 reports the punctual value computed over all the 50 test configurations: it is even more evident that the GPR technique is not able to generalize to test data, contrary to PODNet (and also POD-ANN) which keeps a lower discrepancy between test and train data.

Finally, Figure 9 and 10 illustrate the prediction on two testing parametric points. Also such graphical proof demonstrates a similar outcome using the two best approaches (PODNet and POD-ANN), but we highlight again the crucial difference between them: POD-ANN only returns the predicted

Table 1: Mean errors computed using the Graetz dataset, using POD-GPR, POD-ANN and PODNet models.

|  | POD-GPR | POD-ANN | PODNet |
|---|---|---|---|
| Train mean error | $5.95 \times 10^{-7}$ | 0.1960 | 0.1577 |
| Test mean error | 0.80 | 0.3108 | 0.244 |

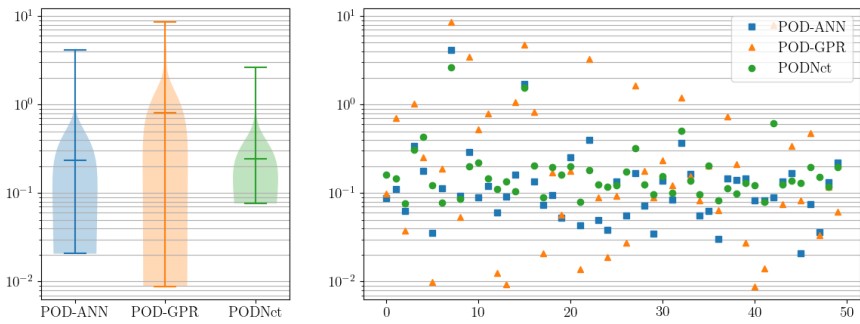

Figure 8: *Graetz Problem*: the 4 modes computed using standard POD approach and using PODNet.

field on the same space where the snapshots are collected, in this case, the finite element triangulation. PODNet instead can predict the output in any coordinate belonging to the spatial domain, making it possible to train the model only using triangulated snapshots and infer it on a pixel grid (as shown in the bottom-left charts).

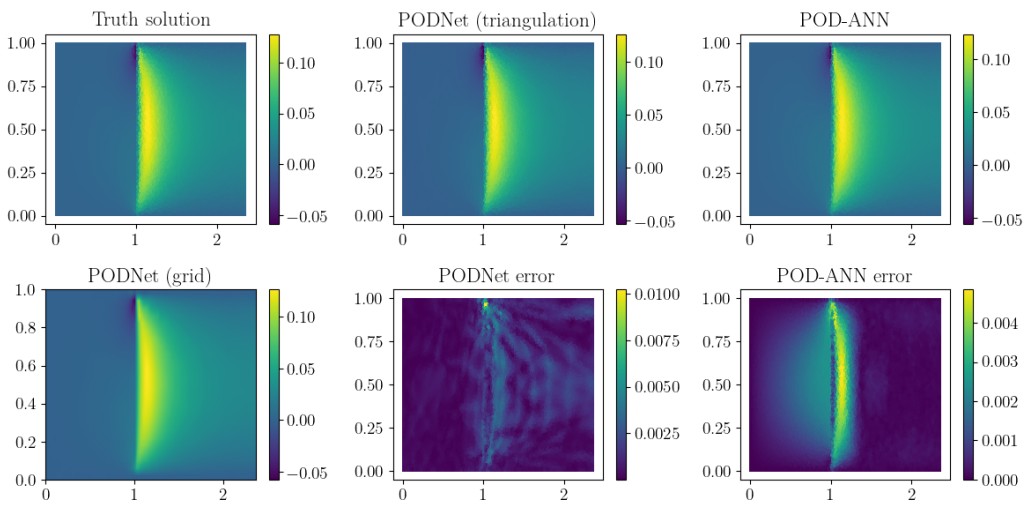

Figure 9: *Graetz Problem*: prediction for $\mu = (7.2239, 9.8438)$.

## 4 CONCLUSION AND NEW PERSPECTIVES

In this study, we have successfully applied the DeepONet architecture to represent Proper Orthogonal Decomposition (POD) at a continuous level. Our initial experiments have demonstrated the architecture's ability to emulate the behavior of the traditional algebraic approach. To enforce or-

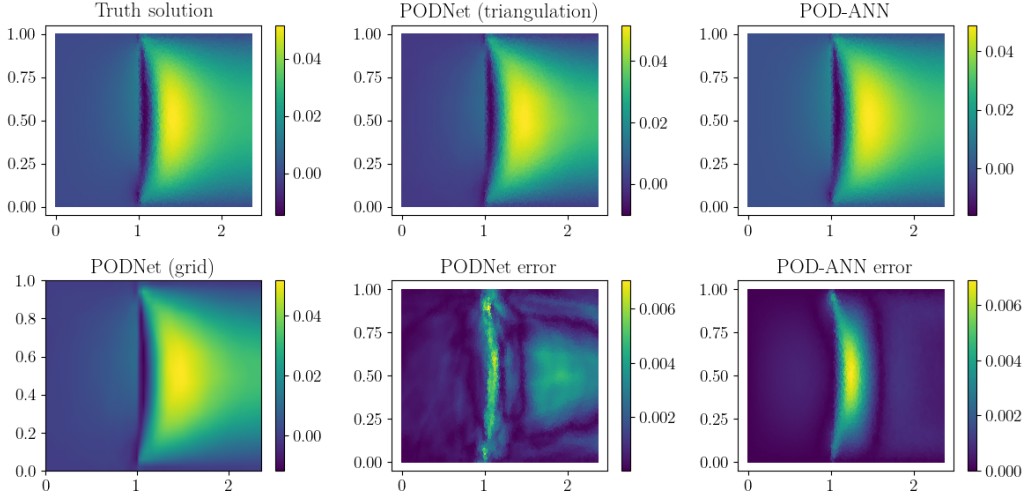

Figure 10: *Graetz Problem*: prediction for $\mu = (3.7401, 2.5090)$.

thogonality, a weak constraint is applied during training, although fine-tuning the associated weight remains a manual and somewhat tedious process, varying from one problem to another.

The central contribution of this work lies in the integration of POD into a machine learning framework. This opens the door to several potential extensions. Firstly, the incorporation of a physics-informed paradigm can enable the extraction of POD modes without the need for extensive data. Additionally, the continuous representation we've developed can facilitate the use of POD with numerical simulations from various sources or spaces, expanding its applicability.

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
