# OpenReview forum: "A space-continuous implementation of Proper Orthogonal Decomposition by means of Neural Networks"
_ICLR.cc/2024/Conference — ICLR 2024 Conference Withdrawn Submission_

### Official Review · Reviewer_tLNX · 2023-11-01

**Soundness:** 3 good
**Presentation:** 3 good
**Contribution:** 1 poor
**Rating:** 5
**Confidence:** 2

**Summary:**

Proper orthogonal decomposition is a method that decomposes spatial-parametric data into spatial structures (features) and parametric structures.  The paper extends POD using a neural network (DeepONet) to emulate the POD structure.

**Strengths:**

The presentation is reasonably clear (but see also below) and the experiments show promise.

**Weaknesses:**

The theoretical background seemed somewhat lacking and the experiments, while promising, consisted of just two problems.

**Questions:**

.

---

### Official Review · Reviewer_zc9S · 2023-11-01

**Soundness:** 2 fair
**Presentation:** 1 poor
**Contribution:** 2 fair
**Rating:** 3
**Confidence:** 3

**Summary:**

The decomposition of a function into a basis of infinite orthonormal functions can be approximated by k < \infty basis functions computed through domain discretization and singular value decomposition (SVD).

In this work, the k basis functions are approximated (learned) by k neural networks (the mode-networks) and the coefficients of the expansion are learned by a different set of k networks (the coefficient-networks), using a previously reported framework (the DeepONet, Lu et al. 21), where the result of two sets of networks is multiplied and summed as in a expansion such as the considered decomposition.
The functions approximated in this multiple-net architecture (DeepONet) have two properties: (1) they are continuous in the spatial domain and the domain of parameters, so they can sample the domains in arbitrary ways (as opposed to the discrete functions given by SVD), and (2) the usual regularization can be imposed in the training to enforce orthogonality in the functions.

**Strengths:**

* Learning continuous functions (using DeepONet) allows arbitrary sampling of the spatial and parameter domains. This can be useful in a variety of applications.

**Weaknesses:**

* The application of networks for function approximation is not a new idea, and it is implemented using a framework (DeepONet) that was already available.

* The constraint to enforce orthogonality is the usual one, it has been applied many times before, and hence has not major interest. Moreover, no hint is given on how to set the relative weight of this regularization term in the cost function.

* The experimental illustration is poor: details and/or references are missing, and there is no justification of the relevance of the examples chosen for the illustrations.

**Questions:**

* Only the first two items (out of four) given in the introduction (page 2) as benefits of the proposed method are actually illustrated in the experiments.

* Section 3 starts by saying that the proposed method, PODNet, will be compared to other alternatives (POD-RBF and POD-ANN), but these (more standard?) alternatives are not explained nor any reference is given about them. In this way one cannot assess the eventual benefits of the proposed method.

* Despite anouncing POD-RBF and POD-ANN, the first experiment does not show the result of POD-ANN.

* The first experiment also shows the "POD basis" result, or "standard POD" result. Is this the one computed using SVD?.

* The first experiment is not very illustrative because the (suposedly beneficial) orthogonality regularization leads to basis functions which are very different from the true basis of the signal. In that case, better reproduction of the true basis is obtained if the orthogonality is not enforced. Well, maybe you can argue that by controlling this term one can decide to enforce (or not!) the orthogonality depending on the problem (as opposed to the SVD, which always leads to orthogonal vectors in the nondiagonal unitary matrices). In any case, no hint is given on how much (or when) to enforce this.

* In the first experiment (k=1 example) the authors talk about "the inverse" of the POD mode. It is not the inverse, it is the POD mode multiplied by a negative factor.

* In the second experiment instead of POD-RBF, the authors mention POD-GPR but no explanation/citation is given at all (not even the meaning of GPR).

* The second experiment shows the result of POD-ANN, but no explanation (nor citation) is given on how it works.

* The second experiment shows experiments of PODNet with different samplings of the spatial domain (grid and triangulation), but when it comes to graphical representation of the error is not clear to which one it corresponds.

* Figures are poorly referenced from the text: in the last paragraph of page 4 there is a missing reference to Fig. 5 (I guess is Fig.5). And Fig. 8 is never cited from the text. What is represented in Fig. 8? Errors?.

---

### Official Review · Reviewer_GGNn · 2023-11-04

**Soundness:** 2 fair
**Presentation:** 2 fair
**Contribution:** 2 fair
**Rating:** 5
**Confidence:** 2

**Summary:**

The paper introduces a novel approach called PODNet, which aims to replicate the capabilities of Proper Orthogonal Decomposition (POD) using neural networks. POD is a widely adopted technique for reduced-order modeling in parametric partial differential equations. The paper presents an innovative fusion of neural networks and traditional reduced-order modeling techniques, providing a continuous representation of spatial modes. While acknowledging that the accuracy gap between neural networks and linear algebraic tools is evident, the authors highlight several advantages, including the ability to work with solutions generated through different discretization schemes and the adaptability of the PODNet framework.

**Strengths:**

Innovative Fusion: The paper's fusion of neural networks and traditional POD techniques to create PODNet is innovative and offers a fresh perspective on reduced-order modeling in PDEs.

Versatility: The ability of PODNet to handle solutions from diverse spatial domains, including numerical solutions from different sources and experimental data, is a significant strength. This versatility can be valuable in real-world applications.

Continuous Modes: The continuous representation of spatial modes enables PODNet to provide predictions for unseen parameters and coordinates without the need for additional interpolation, which is a notable advantage.

Generalization and Stability: The cohesive approach of simultaneously optimizing the mapping to the reduced space and the approximation of the solution manifold enhances generalization and mitigates potential stability issues.

**Weaknesses:**

Accuracy Gap: The paper acknowledges that there is still an accuracy gap between neural networks and traditional POD methods. It would be beneficial to discuss strategies or scenarios where this gap could be narrowed or addressed.

Complexity: While the versatility of PODNet is a strength, it may introduce complexity in the implementation and practical application. More insights into potential challenges and complexities would be helpful.

Comparative Analysis: The paper mentions that several extensions of POD have been proposed. It would be valuable to include a comparative analysis with some of these extensions to provide a clear understanding of how PODNet compares to existing approaches.

**Questions:**

Can you provide more details on the potential strategies or research directions aimed at narrowing the accuracy gap between neural networks and traditional POD methods when using the PODNet approach?

In practical applications, how does the continuous representation of spatial modes affect the computational efficiency and accuracy of predictions, especially when handling complex PDEs?

Are there specific examples of real-world problems or domains where the versatility of PODNet, which can handle diverse data sources, has shown significant advantages over traditional POD methods?

Could you elaborate on the extensions of POD that can be seamlessly transferred to the PODNet approach, and how these extensions can enhance its capabilities?

Considering that neural networks may require substantial computational resources for training, what are the computational implications of implementing the PODNet approach in terms of both training and inference?

---

### Official Review · Reviewer_xumW · 2023-11-08

**Soundness:** 1 poor
**Presentation:** 2 fair
**Contribution:** 1 poor
**Rating:** 3
**Confidence:** 4

**Summary:**

This work targets the proper orthogonal decomposition for reduced-order modeling of physical governing functions. As an extension of the DeepONet, a PODNet, consisting of multiple neural networks, is proposed to approximate the modes and corresponding coefficients, respectively.

**Strengths:**

The motivation of the proposed method sounds adequate for this topic.

**Weaknesses:**

The proposed structure could be useful, but the testing cases are too limited, e.g., low-dimension problems (2) with small numbers of basis functions (1-4). Also, the baseline is too naïve as more SOTA should be compared with. The reviewer doubts its effectiveness and generalizability.

Based on the claimed contributions in the Introduction, this paper seems to be an incomplete work to study the integration of neural network approximation with proper orthogonal decomposition. Especially the embedding with a physics-informed neural network is essential and interesting, a good direction to move forward.

**Questions:**

If the reviewer understands correctly, each basis function is represented by a neural network, so as each corresponding coefficient. It could introduce high computation. Have you considered the accuracy-computation trade-off?

Page 4 bottom: Figure ??

---

### Official Review · Reviewer_7r7x · 2023-11-10

**Soundness:** 2 fair
**Presentation:** 1 poor
**Contribution:** 2 fair
**Rating:** 3
**Confidence:** 4

**Summary:**

The authors present PODNet, a method to approximate Proper Orthogonal Decomposition (POD) using neural networks. The method leverages two types of neural networks, one for the spatial coordinates and one for the parameters; the dot product between their outputs results in the prediction.
The authors evaluate the proposed approach on two simulated examples.

**Strengths:**

Overall the approach makes sense. The orthogonality constraint is well designed.
The approach yields positive results on the simulated examples, at least up until test time = 20.

The modes recovered on the simulated examples are very promising.

**Weaknesses:**

### Details of the approach
“or this experiment, the ModeNets are multi-layer perceptron (MLP) composed of 3 hidden layers of 25, 15, and 10 neurons. The activation function is the hyperbolic tangent. The CoefficientNet is still an MLP, but with 18, 18, and 8 neurons in the three hidden layers, and applying Softplus as the activation function.”
What is the justification for these rather uncommon choices?
One would have typically gone for ReLU activation functions and a deeper network, as they would be more flexible function approximators. Also what is the motivation for using Adam first and then LBGFS?
It would be worthwhile to share the ablation studies done to arrive to this selection, to assess how sensitive the method is to these choices.

Same question for the Graetz problem network and training. Does this mean that one would need to hand pick the parameters of the ModeNets (and the orthogonality coefficient, and the number of epochs of each optimiser) for every new problem? How should one approach this?


### Evaluation
I understand that the main advantage of the proposed method is to improve on the inference speed. As such, in addition to a comparison of the modes recovered by PODNet and POD, it would be very relevant to have a comparison of the inference time.

It would be helpful to provide std deviations for the training table.


### Writing
The paper writing could be significantly improved which would be of great help to readers.
There are many tools to help one do so, it’s worth using them.

Make the Figure 1 bigger, so as to have the same font size as the text.

Small comments:
“reduced-order modeling somehow anchored” -> remove “somehow”
“but the model can be surely trained also using different metrics” -> “but the formulation is general and the model can be trained using other loss” or something to this effect
“the evaluation ad the” -> “at the”

“The idea to efficiently compute the orthogonality coefficient between the modes is to evaluate such networks”: poor wording
“helping the training convergence since the weak imposition of the orthogonality”: replace “since” by “with”

“Figure ?? shows” -> fix reference
In the text Figure 4 shows the error for k=1 but the caption of figure 4 says k=2.

**Questions:**

### Evaluation
How does the performance change when changing the number of training points?
Why select only 150 training point in the Graetz Problem?

Why only 50 test points? I believe that with such a number there would be a significant noise coming simply from the test points being sampled.